# Trichostatin D as a Novel KLF2 Activator Attenuates TNFα-Induced Endothelial Inflammation

**DOI:** 10.3390/ijms232113477

**Published:** 2022-11-03

**Authors:** Lijuan Lei, Minghua Chen, Chenyin Wang, Xinhai Jiang, Yinghong Li, Weizhi Wang, Shunwang Li, Liping Zhao, Ren Sheng, Jiangxue Han, Yuyan Zhang, Yuchuan Chen, Biying Yan, Yexiang Wu, Liyan Yu, Shuyi Si, Yanni Xu

**Affiliations:** NHC Key Laboratory of Biotechnology of Antibiotics, National Center for Screening Novel Microbial Drugs, Institute of Medicinal Biotechnology, Chinese Academy of Medical Sciences & Peking Union Medical College (CAMS & PUMC), 1# Tiantan Xili, Beijing 100050, China

**Keywords:** trichostatin D, KLF2, endothelial inflammation, atherosclerosis, HDAC, NLRP3, VCAM-1

## Abstract

Krüppel-like factor 2 (KLF2) is an atherosclerotic protective transcription factor that maintains endothelial cell homeostasis through its anti-inflammatory, anti-oxidant, and antithrombotic properties. The aim of this study was to discover KLF2 activators from microbial secondary metabolites and explore their potential molecular mechanisms. By using a high-throughput screening model based on a *KLF2* promoter luciferase reporter assay, column chromatography, electrospray ionization mass spectrometry (ESI-MS), and nuclear magnetic resonance (NMR) spectra, trichostatin D (TSD) was isolated from the rice fermentation of *Streptomyces* sp. CPCC203909 and identified as a novel KLF2 activator. Real-time-quantitative polymerase chain reaction (RT-qPCR) results showed that TSD upregulated the mRNA level of *KLF2* in endothelial cells. Functional assays showed that TSD attenuated monocyte adhesion to endothelial cells, decreased vascular cell adhesion protein 1 (VCAM-1) and intercellular adhesion molecule 1 (ICAM-1) expression, and exhibited an anti-inflammatory effect in tumor necrosis factor alpha (TNFα)-induced endothelial cells. We further demonstrated through siRNA and western blot assays that the effects of TSD on monocyte adhesion and inflammation in endothelial cells were partly dependent on upregulating KLF2 expression and then inhibiting the NOD-like receptor protein 3 (NLRP3)/Caspase-1/interleukin-1beta (IL-1β) signaling pathway. Furthermore, histone deacetylase (HDAC) overexpression and molecular docking analysis results showed that TSD upregulated KLF2 expression by inhibiting HDAC 4, 5, and 7 activities. Taken together, TSD was isolated from the fermentation of *Streptomyces* sp. CPCC203909 and first reported as a potential activator of KLF2 in this study. Furthermore, TSD upregulated KLF2 expression by inhibiting HDAC 4, 5, and 7 and attenuated endothelial inflammation via regulation of the KLF2/NLRP3/Caspase-1/IL-1β signaling pathway.

## 1. Introduction

Atherosclerosis is a chronic inflammatory disease caused by lipid metabolism disorder and is the pathological basis of many cardiovascular diseases, such as coronary heart disease and myocardial infarction [1,2,3]. Endothelial cells play an important role in vascular homeostasis, thrombosis, inflammation, and other processes [4,5]. Endothelial dysfunction caused by endothelial cell damage is an important initiating event in the occurrence and development of atherosclerotic plaques. Damaged endothelial cells secrete a variety of adhesion molecules and chemokines to attract circulating monocytes to adhere to endothelial cells [6,7]. Monocytes migrate to the subcutaneous space through endothelial junctions and differentiate into macrophages, which proceed to take up various forms of modified low-density lipoprotein (LDL), especially oxidized LDL, to form foam cells, an early process in the formation of fat streaks and atherosclerotic plaques [8,9].

Krüppel-like factor 2 (KLF2), a member of the Krüppel-like factors (KLFs), is a zinc finger transcription factor. The zinc finger domain of KLFs is located at the C-terminal and plays a role in DNA binding and nuclear localization. The N-terminal region of KLFs is a highly diverse non-DNA binding region that mediates protein-protein interactions to activate or inhibit transcription [10,11,12]. KLF2 is an atherosclerotic protective transcription factor that regulates endothelial cell inflammation, thrombosis, and angiogenesis [4,13,14,15,16]. Therefore, discovery of new small molecule compounds that are KLF2 activators may provide new therapeutic strategies for atherosclerosis.

The NOD-like receptor protein 3 (NLRP3) inflammasome complex is composed of NLRP3, an adaptor protein, apoptosis-associated speck-like protein containing CARD (ASC), and an effector protein, Caspase-1 [17,18]. Through this inflammasome complex, pro-caspase-1 is cleaved into its active isomer, Caspase-1, and pro-interleukin-1beta (pro-IL-1β) is broken down into its active isomer IL-1β [19]. In turn, the resultant release of IL-1β leads to a dysfunctional endothelium and an increase in inflammatory mediators in the form of vascular cell adhesion protein 1 (VCAM-1), which contributes to atherosclerosis [20]. Previous studies showed that simvastatin increased the expression of forkhead box P transcription factor 1 (Foxp1) and KLF2 in endothelial cells and thus inhibited the expression of NLRP3, Caspase-1, and IL-1β and the activation of endothelial cell inflammation [4].

In this study, we obtained *Streptomyces* sp. CPCC 203909, which upregulated *KLF2* luciferase expression, using a previously established high-throughput screening model based on a *KLF2* promoter luciferase reporter assay [21,22]. We isolated an active compound trichostatin D (TSD) from its fermentation products that upregulated *KLF2* mRNA levels, and we demonstrated that TSD alleviated monocyte adhesion and exhibited anti-inflammatory effects in tumor necrosis factor alpha (TNFα)-induced endothelial cells. Furthermore, we demonstrated that TSD upregulated KLF2 expression by inhibiting histone deacetylase (HDAC) 4, 5, and 7 and attenuated endothelial inflammation through regulation of the KLF2/NLRP3/Caspase-1/IL-1β signaling pathway.

## 2. Results

### 2.1. Isolation and Identification of TSD from Streptomyces sp. CPCC 203909

In the preliminary screening assay, we found that the fermentation of *Streptomyces* sp. CPCC 203909 increased the luciferase activity of *KLF2* in COS-7 cells. Following bioassay-guided isolation using silica column chromatography, octadecyl-silica RP flash column chromatography, and semi-preparative HPLC, one positive compound was obtained as a KLF2 activator. A comparison of the ultra violet (UV), electrospray ionization mass spectrometry (ESI-MS), and nuclear magnetic resonance (NMR) data with previous references showed that the structure of the positive compound was TSD (Figure 1B) [23].

The TSD was a white amorphous powder. The molecular formula of TSD was established as C_23_H_32_N_2_O_8_ on the basis of ESI-MS data ([M + H]^+^ m/z 465.2). The ^1^H NMR (600 MHz, CD_3_OD-d4) spectrum exhibited signals (Figure 1A): a para-disubstituted benzene ring at δ_H_ 6.73 (2H, d, *J* = 9.0 Hz, H-10,12) and 7.87 (2H, d, *J* = 9.0 Hz, H-9,13), a *trans*-olefin bond at δ_H_ 5.88 (1H, d, *J* = 15.0 Hz, H-2) and 7.25 (1H, d, *J* = 15.6 Hz, H-3), between an olefinic proton δ_H_ 5.97 (1H, d, *J* = 9.0 Hz, H-5) and 1.28 (3H, d, *J* = 6.6 Hz, 6-Me) through a methine proton at δ_H_ 4.56 (1H, dq, *J* = 9.6, 7.2 Hz, H-10,12), a methyl group at δ_H_ 1.94 (3H, s, 4-Me) and two equivalent methyl groups at δ_H_ 3.06 (6H, s, N-Me_2_). In addition, a glucose group at δ_H_ 3.34 (1H, dd, *J* = 9.6, 3.0 Hz, H-2′), δ_H_ 3.66 (1H, t, *J* = 9.0 Hz, H-3′), δ_H_ 3.35 (1H, t, *J* = 9.0 Hz, H-4′), δ_H_ 3.47 (1H, m, H-5′), δ_H_ 3.86 (1H, d, *J* = 12.0 Hz, H-6′), and δ_H_ 3.72 (1H, dd, *J* = 11.4, 3.0 Hz, H-6′) was also observed in the ^1^H NMR spectrum. The ^13^C NMR (600 MHz, CD_3_OD-d4) spectrum exhibited signals: quaternary carbons at δ_C_ 201.5 (C-7), δ_C_ 124.8 (C-8), δ_C_ 155.6 (C-11) and an olefinic quaternary carbon at δ_C_ 134.6 (C-4), an olefinic carbon at δ_C_ 116.5 (C-2), δ_C_ 147.4(C-3), δ_C_ 142.5 (C-5), primary carbons at δ_C_ 12.8 (4-Me), δ_C_ 18.5 (6-Me), δ_C_ 40.2 (N-Me_2_), and tertiary carbons at δ_C_ 41.9 (C-6), δ_C_ 132.1 (C-9,13), δ_C_ 112.1 (C-10,12). In addition, a glucose group at δ_C_ 105.7 (C-1′), δ_C_ 73.4 (C-2′), δ_C_ 74.9 (C-3′), δ_C_ 71.4 (C-4′), δ_C_ 75.2 (C-5′), δ_C_ 62.6 (C-6′) was observed in the ^13^C NMR spectrum.

### 2.2. TSD Increases KLF2 Activity and mRNA Levels

To confirm whether TSD increased KLF2 activity, a dose-response curve was generated using *KLF2* promoter luciferase assays in COS-7 cells, as described in the Methods section. It was observed that TSD significantly increased the luciferase activity of *KLF2* by more than 40-fold at 2.5 μM, with an EC_50_ value of 0.15 μM (Figure 1C).

We then evaluated the mRNA level of *KLF2* in human umbilical vein endothelial cell (HUVECs) after treatment with TSD. The real-time-quantitative polymerase chain reaction (RT-qPCR) data indicated that TSD significantly increased *KLF2* mRNA levels at 5 and 10 μM and by more than 4-fold at 10 μM (Figure 1D). In addition, cell viability experiments showed that TSD caused no obvious cytotoxicity at the experimental concentration from 1 μM to 40 μM in HUVECs (Figure 1E).

### 2.3. TSD Attenuated Monocyte Adhesion to HUVECs and Exhibited Anti-Inflammatory Effects

VCAM1 and intercellular adhesion molecule 1 (ICAM-1) are two adhesion molecules secreted after endothelial cell injury that reflect the degree of endothelial cell injury. KLF2 is a potential anti-inflammatory transcription factor that plays an important role in maintaining endothelial cell function by inhibiting the expression of VCAM-1 and ICAM-1 [4,24,25,26].

To evaluate the anti-inflammatory effect of TSD, an endothelial cell adhesion model induced by TNFα was constructed. When compared with the vehicle control (TSD at 0 μM), TNFα significantly induced THP-1 monocyte adhesion to HUVECs, while TSD (5 and 10 μM) treatment significantly attenuated TNFα-induced monocyte adhesion (Figure 2A,B), which indicated that TSD has an anti-inflammatory effect.

The potential molecular mechanism by which TSD reduced monocyte adhesion was investigated. As shown in Figure 2C, TNFα significantly decreased *KLF2* mRNA levels when compared with the vehicle control, while TSD upregulated *KLF2* mRNA levels in TNFα-induced HUVECs when compared with the TNFα-only group. In addition, TNFα significantly increased *VCAM1* and *ICAM1* mRNA levels when compared with the vehicle control, while TSD significantly decreased the mRNA (Figure 2C) and protein levels of VCAM-1 and ICAM-1 (Figure 2E) in the TNFα-treated cells when compared with the TNFα-only group. Taken together, these results indicated that TSD exerted an anti-inflammatory effect on HUVECs.

### 2.4. Anti-Inflammatory Effects of TSD in TNFα-Induced HUVECs Depend on KLF2

TSD upregulated KLF2 expression and exhibited anti-inflammatory effects in endothelial cells. To investigate whether the anti-inflammatory effects of TSD on HUVECs were dependent on KLF2, HUVECs were transfected with *KLF2* siRNA (siKLF2) for 24 h, and then, TNFα-induced monocyte adhesion assays were performed. As shown in Figure 3A, siKLF2 downregulated the mRNA level of *KLF2* to about 30% compared with the siControl group. The same as above, siControl and TNFα-treated group significantly increased monocyte adhesion compared with the siControl group (Figure 3B), and TSD remarkably decreased the TNFα-induced enhancement of monocyte adhesion (Figure 3B). siRNA knockdown of *KLF2* resulted in a higher level of monocyte adhesion compared with the TSD treatment group (Figure 3A–C). In addition, the decreased protein levels of VCAM-1 and ICAM-1 induced by TSD were partially increased when *KLF2* was knocked down (Figure 3D,E). Thus, these data indicate that TSD exerted anti-inflammatory effects on endothelial cells in part by upregulating *KLF2* expression.

### 2.5. TSD Exerts Anti-Inflammatory Effects on Endothelial Cells through the KLF2/NLRP3/Caspase-1/IL-1β Signaling Pathway

KLF2 inhibited activation of the NLRP3 inflammasome [17,18,27], and inhibition of NLRP3 expression reversed the inflammation of endothelial cells [28,29,30]. To investigate whether TSD affected NLRP3, Casapase-1, and IL-1β protein levels, western blot experiments were performed. TNFα stimulation significantly increased NLRP3, Casapase-1, and IL-1β protein levels, while TSD reversed this increase in a dose-dependent manner (Figure 4A,B). Furthermore, by knocking down *KLF2* with siRNA, the inhibitory effect of TSD on NLRP3, Casapase-1, and IL-1β proteins was reversed (Figure 4C,D), which indicated that the TSD-mediated inhibition of the NLRP3/Caspase-1/IL-1β signaling pathway was partially dependent on KLF2.

### 2.6. TSD Upregulates the Expression of KLF2 by Inhibiting HDAC4, HDAC5, and HDAC7

Histone deacetylases catalyze DNA deacetylation and play a crucial role in the modulation of atherosclerosis [31]. TSD is an analog of trichostatin A (TSA), which is a pan-HDAC inhibitor [32,33,34]. Studies have found that inhibition of HDAC activity induced KLF2 expression, which then contributed to reducing inflammation of endothelial cells [35,36]. We wondered whether the upregulation of KLF2 induced by TSD was related to the inhibition of a subtype of HDACs.

To investigate whether TSD upregulated KLF2 and exerted anti-inflammatory effects through a subtype of HDACs, we first transfected eight HDAC plasmids into COS-7 cells. Western blot experiments showed that the HDAC1-8 plasmids were successfully transfected and expressed in COS-7 cells. To examine which HDAC type was related to KLF2 expression, we co-transfected HDAC1-8 expression plasmids and KLF2 luciferase reporter plasmids in COS-7 cells, to detect which HDAC subtype acted on *KLF2* according to the luciferase activity. As shown in Figure 5A, HDAC4, HDAC5, and HDAC7 inhibited *KLF2* transcriptional activity. We then selected HDAC4, HDAC5, and HDAC7 plasmids to examine the effects of TSD on these three subtype HDACs using a *KLF2* upregulating screening model. The results showed that TSD reversed *KLF2* transcriptional activity inhibited by HDAC4, HDAC5, and HDAC7 (Figure 5B), which indicated that TSD had an inhibitory effect on HDAC4, HDAC5, and HDAC7.

### 2.7. Interaction Modes between TSD and HDAC4, HDAC5, or HDAC7

To investigate the interaction modes between TSD and HDAC4, HDAC5, or HDAC7, molecular docking analysis between HDAC4 (PDB code: 4A69), HDAC5 (PDB code: 5UWI), and HDAC7 (PDB code: 3C10) proteins with TSD was performed using Discovery Studio 4.5 software (NeoTrident Technology Ltd., Beijing, China). The Libdock scores of TSD with HDAC4, HDAC5, and HDAC7 proteins were 138.2, 154.6, and 134.8, respectively. The residues Asp196, His198, and Asp290 likely contributed to the van der Waals force between TSD and the HDAC4 protein (Figure 6A); the residues Asp18, Thr21, Gly22, Lys23, and Thr42 likely contributed to the van der Waals force between TSD and the HDAC5 protein (Figure 6B); and the residues Asp707, His709, and Asp801 likely contributed to the van der Waals force between TSD and the HDAC7 protein (Figure 6).

In conclusion, TSD upregulated KLF2 expression by inhibiting HDAC4, HDAC5, and HDAC7 and alleviated endothelial inflammation through regulation of the KLF2-dependent NLRP3/Caspase-1/IL-1β signaling pathway (Figure 7).

## 3. Discussion

Endothelial cells regulate vascular tone, endothelium integrity, thrombosis, endothelial injury, and repair [4,5,6,7]. Pro-inflammatory cytokines such as TNFα and IL-1β can trigger endothelial cell inflammation. In response to injury, endothelial cells are activated and produce VCAM-1, ICAM-1, E-selectin, and other molecules that lead to the adhesion of monocytes and neutrophils to endothelial cells [37,38]. It was found that the activation of KLF2 inhibited the expression of VCAM-1 and E-selectin, thereby exerting anti-inflammatory effects [39,40,41]. Therefore, the discovery of KLF2 activators in endothelial cells might prevent and treat cardiovascular diseases associated with endothelial dysfunction, such as atherosclerosis.

In this study, using a high-throughput drug screening model based on a *KLF2* luciferase reporter gene, the fermentation products of *Streptomyces* sp. CPCC 203909 were isolated and purified, and the active compound TSD was obtained. According to a literature search, the regulatory effect of TSD on *KLF2* transcription factor expression was identified for the first time. Our previous study reported that TSD had a positive role in anti-atherosclerosis by upregulating the activity of CD36 and lysosomal integral membrane protein II analogous-1 (CLA-1), which is important in cholesterol efflux. The results in this study showed that TSD reversed TNFα-induced monocyte adhesion, upregulated mRNA levels of *KLF2* in endothelial cells, and downregulated mRNA and protein levels of the inflammatory adhesion molecules VCAM-1 and ICAM-1 in TNFα-induced endothelial cells. We also demonstrated that the anti-inflammatory effects of TSD on endothelial cells were partially dependent on KLF2. In addition, it was demonstrated that TSD inhibited the expression of the NLRP3/Caspase-1/IL-1β signaling pathway by upregulating KLF2 expression, thereby exerting an anti-inflammatory effect.

TSD is a structural analog of TSA. TSA is a known pan-inhibitor of HDACs that thereby regulating the expression of related genes [32,33,34]. It was reported that HDAC4, 5, and 7 inhibited KLF2 expression, which in turn had an important role in anti-inflammation, sepsis, and angiogenesis [16,35,42,43,44]. TSA prevented the up-regulation of the endothelial dysfunction markers (VCAM-1 and ICAM-1) [45,46,47]. Our study found that TSD inhibited the activity of HDAC4, HDAC5, and HDAC7 and then upregulated the expression level of KLF2.

In addition, as statins are studied well and proven to attenuate endothelial dysfunction via KLF2 activation [4,27], we then performed *KLF2* promoter luciferase assays in COS-7 cells and TNFα-induced monocyte adhesion assays to compare the effects of statins (Simvastatin and Atorvastatin) with TSD. As shown in Appendix A, Simvastatin maximally increased the luciferase activity of KLF2 by about 8-fold, with an EC_50_ value of 1.74 μM (Appendix A), and Atorvastatin maximally increased the luciferase activity of KLF2 by about 12-fold, with an EC_50_ value of 5.14 μM (Appendix A). Monocyte adhesion assays results showed that Simvastatin (1 and 10 μM) and Atorvastatin (1 and 10 μM) treatment attenuated TNFα-induced monocyte adhesion (Appendix A) and exhibited anti-inflammatory effects in endothelial cells (Appendix A). Furthermore, our results indicated that TSD is more effective in upregulating KLF2 expression and anti-inflammatory effects than Simvastatin and Atorvastatin according to the results of *KLF2* promoter luciferase assay and inhibiting monocyte adhesion (Appendix A).

In conclusion, our work found that TSD is a novel KLF2 activator and that it has a strong anti-inflammatory effect on endothelial cells. In the future, the anti-atherosclerotic effect of TSD will be studied in western diet-induced apolipoprotein E knockout mice.

## 4. Materials and Methods

### 4.1. Fermentation of Streptomyces sp. CPCC 203909

*Streptomyces* sp. CPCC203909 was deposited in the China Pharmaceutical Culture Collection (Institute of Medicinal Biotechnology, Chinese Academy of Medical Sciences (CAMS) and Peking Union Medical College (PUMC), No. CPCC 203909). *Streptomyces* sp. CPCC203909 was cultured on slants of BD Difco ISP Medium 2 (0.4% yeast extract, 1.0% malt extract, 0.4% dextrose, and 2.0% agar; Becton, Dickinson and Company, New Jersey, USA) at 28 °C. Colonies were picked with a plastic inoculating loop and inoculated into 3 mL of A2 liquid medium (1.0% glucose, 3.0% amylogen, 2.0% cottonseed meal, 0.3% yeast extract, 0.3% ammonia sulfate, 0.1% magnesium sulfate, 0.1% dipotassium hydrogen phosphate, 0.1% sodium chloride, and 0.5% calcium carbonate), and the final pH was adjusted to 7.2. Flasks with the inoculated media were incubated at 28 °C on a rotary shaker at 180 rpm for 2 days after sterilization. For the seed culture, the culture was added to 100 mL of A2 liquid medium (at a ratio of 1:100) on a rotary shaker at 180 rpm at 28 °C and incubated for 2 days. Next, two hundred and fifty fermentation bags containing 50 g of rice and 50 mL of distilled water were used for fermentation. The contents were soaked and autoclaved at 121 °C for 30 min. After cooling to room temperature, 5.0 mL of the seed medium was added to each bag at 28 °C for 30 days.

### 4.2. Extraction, Isolation, Purification, and Structure Identification of Trichostatin D

After a 30-day fermentation, the fermentation products were soaked with ethanol overnight, ultrasonically extracted for 30 min, and the liquid was collected by filtration. The filtered liquid was then extracted with ethyl acetate. The ethyl acetate organic solvent phase was evaporated under vacuum with a rotary evaporator until a crude extract (111.36 g) was obtained. The extract was fractionated by silica column chromatography with dichloromethane (CH_2_Cl_2_) and methanol (MeOH) (at a ratio of 50:1, 30:1, 10:1, 5:1 and MeOH). The fraction in MeOH yielded four fractions, L1–L4. Fraction L1 (1.72 g) was further separated by a reversed-phase (RP) flash column chromatography and eluted by 35% acetonitrile (ACN) in distilled H_2_O at 5 mL/min to provide seven sub-fractions (L1-1–L1-7). Fraction L1-3 was purified by semi-preparative RP high-performance liquid chromatography (HPLC; 30% ACN in H_2_O (containing 0.1% trifluoroacetic acid)), 1 mL/min, UV detection at 254 nm; column: X Charge C18 5 μm, Acchrom, Beijing, China) to isolate TSD (32.43 mg, retention time = 39.6 min).

TSD was determined by UV spectrometry (SHIMADZU, Kyoto, Japan), ESI-MS (Agilent Technologies, Santa Clara, CA, USA), and ^1^H-NMR and ^13^C-NMR spectrometry (600 MHz, Agilent Technologies, Santa Clara, CA, USA).

### 4.3. Cell Culture

The COS-7 cell line was obtained from Cell Resource Center (Institute of Basic Medical Sciences, CAMS, Beijing, China) and was cultured in Dulbecco’s modified Eagle medium (DMEM) (Thermo Fisher Scientific, Waltham, MA, USA) with 10% fetal bovine serum (FBS) (Thermo Fisher Scientific, Waltham, MA, USA). HUVECs was obtained from PromoCell (Heidelberg, Germany) and was grown in endothelial cell medium (ECM) (ScienCell). The THP-1 cells line was provided by Cell Resource Center (Institute of Basic Medical Sciences, CAMS, Beijing, China) and was cultured in RPMI 1640 (Thermo Fisher Scientific, Waltham, MA, USA) containing 10% FBS. All cells were cultured at 37 °C with 5% CO_2_ in a cell incubator.

### 4.4. Cell Transfection and Luciferase Assay

*KLF2* luciferase assays were performed as previously described [21,22]. KLF2-luc promoter-driven luciferase reporter plasmids (KLF2-luc) were gifted from Prof. Mukesh Jain. COS-7 cells plated in a 96-well plate were transfected with 0.2 μg per well of KLF2-luc plasmids using Lipofectamine 2000 (Thermo Fisher Scientific, Waltham, MA, USA) in Opti-MEM (Thermo Fisher Scientific, Waltham, MA, USA) for 6 h. TSD diluted to different concentrations (0, 0.001, 0.01, 0.1, 2.5, 5, 10, 20, and 40 μM) in 200 μL DMEM containing 10% FBS was added to a 96-well plate and incubated for 24 h. Simvastatin (#T0687, topscience, shanghai, China) and Atorvastatin (#T3116, topscience, shanghai, China) diluted to different concentrations (0.001, 0.01, 0.1, 1.0, 5, 10, 20, 40, and 80 μM) were added to COS-7 cells transfected with a KLF2-Luc plasmid for 24 h. The activity of the *KLF2* luciferase reporter gene was detected using a Bright-Lite Luciferase Assay System (Vazyme, Nanjing, China) with a microplate reader (Perkin Elmer, MA, USA). The relative upregulation of the *KLF2* luciferase activity was normalized to the vehicle (0.1% DMSO).

For HDAC inhibition assays, COS-7 cells plated in a 96-well plate were transfected with 0.2 μg per well of HDAC Flag plasmids (HDAC1, #13820; HDAC2, #36830; HDAC3, #13819; HDAC4, #13821; HDAC5, #13822; HDAC6, #13823; HDAC7, #13824; and HDAC8, #13825; Addgene, Cambridge, MA, USA) for 6 h. Then, the cells were treated with TSD, as described in Section 2.4.

### 4.5. Cell Viability Assay

HUVECs were inoculated in a 96-well plate, and TSD at different concentrations (0, 1, 2.5, 5, 10, 20, and 40 μM) was added. The cells were incubated for 24 h at 37 °C with 5% CO_2_. A sensitive colorimetric assay was performed to quantify the viable cells in cytotoxicity assay with a Cell Counting Kit-8 (CCK-8, Applygen, Beijing, China) using a microplate reader. For the assay, 10 µL of the CCK-8 working solution was added to each well, and then cells were incubated at 37 °C for 2 h. The absorbance was measured at 450 nm.

### 4.6. RNA Isolation and RT-qPCR

The cells were grown in 6-well plates and treated with different concentrations of TSD (0.1, 1, 5, and 10 μM) for 24 h. Total RNA from the cells was extracted using a QIAGEN RNeasy Mini kit (Qiagen, Hilden, Germany), then converted into complementary DNA (cDNA) using a TransScript One-Step gDNA Removal kit, and the cDNA was synthesized using Synthesis SuperMix (Transgen Biotech, Beijing, China). RT-qPCR was performed in an FTC-3000 Real-Time Quantitative Thermal Cycler (Funglyn Biotech Inc., Richmond Hill, ON, Canada) using FastStart Universal SYBR Green Master Mix (Roche, Basel, Switzerland) and the corresponding primers. Relative mRNA expression of the target genes was normalized to *ACTB* (encoding β-actin), and the results were quantitated. Sequences of the specific human primers were as follows: human *KLF2* (forward: 5′-TGGAGGCCAAGCCAAAG-3′, reverse: 5′-CGAACTCTTGGTGTAGGTCTTG-3′); human *VCAM1* (forward: 5′-GATTGGTGACTCCGTCTCATT-3′, reverse: 5′-CCTTCCCATTCAGTGGACTATC-3′); human inter-cellular adhesion molecule 1 (*ICAM1*) (forward: 5′-GTAGCAGCCGCAGTCATAAT-3′, reverse: 5′-GGGCCTGTTGTAGTCTGTATTT-3′); and human *ACTB* (forward: 5′-GGACCTGACTGACTACCTCAT-3′, reverse: 5′-CGTAGCACAGCTTCTCCTTAAT-3′).

### 4.7. Western Blot Analysis

The total protein from cells was extracted using RIPA buffer (Solarbio, Beijing, China) supplemented with 1 mmol/L phenylmethanesulfonyl fluoride (PMSF) (Solarbio, Beijing, China) for 30 min at 4 °C. Then, the cell lysate was centrifuged (12,000 rpm) at 4 °C for 30 min to collect the protein extracted in the supernatant. The protein concentrations were determined with bicinchoninic acid (BCA) Protein Assay kits (Thermo Fisher Scientific, Waltham, MA, USA). The total cell proteins were separated by SDS-PAGE gels and transferred to polyvinylidene fluoride (PVDF) membranes. Next, the membranes were blocked with 5% skim milk (Becton, Dickinson and Company, New Jersey, USA) for 1 h at room temperature. Then, the membranes were incubated with appropriate primary antibodies overnight at 4 °C, followed by incubation with appropriate secondary anti-rabbit and anti-mouse antibodies (Cell Signaling Technology, Danvers, MA, USA) at room temperature for 1 h. All relative protein levels of target genes were normalized to β-actin or β-tubulin. Densitometric analysis of the blots was carried out with NIH ImageJ software. The primary antibodies included human VCAM-1 (Abcam, Cambridge, UK), human ICAM-1 (Cell Signaling Technology, Danvers, MA, USA), human NLRP3 (Cell Signaling Technology, Danvers, MA, USA), human Caspase-1 (Abcam, Cambridge, UK), human IL-1β (Abcam, Cambridge, UK), human Flag (Applygen, Beijing, China), human β-actin (Proteintech, Beijing, China), and human β-Tubulin (Proteintech, Beijing, China).

### 4.8. Monocyte Adhesion Assay

The monocyte adhesion assay was performed as previously described [21,22]. HUVECs were plated in 6-well plates and incubated with the vehicle (0.1% DMSO), TSD at different concentrations (0.1 μM, 1 μM, 5 μM, and 10 μM), Simvastatin (1 μM and 10 μM) or Atorvastatin (1 μM and 10 μM) in a CO_2_ incubator for 18 h. Recombinant TNFα (R&D Systems, Minneapolis, MN, USA) at a final concentration of 10 ng/mL was then added to the culture for another 6 h. Next, 0.5 mL of THP-1 cells at a density of 4 × 10^6^/mL was added to the culture and co-incubated for 30 min. Then, the cells were gently washed three times with FBS-free medium to remove non-adherent THP-1 cells. Photographs of cells were taken under a microscope (Leica CM1950, Wetzlar and Mannheim, Germany).

### 4.9. Small Interfering RNA (SiRNA) Assay

HUVECs were cultured in 6-well cell culture plates and transfected with siControl (50 nM; Santa Cruz Biotechnology, Santa Cruz, CA, USA) or siKLF2 (50 nM; Santa Cruz Biotechnology, Santa Cruz, CA, USA) with Lipofectamine RNAi^MAX^ (Thermo Fisher Scientific, Waltham, MA, USA) in Opti-MEM (Thermo Fisher Scientific, Waltham, MA, USA). After 6 h, the medium was replaced with ECM and incubated for another 18 h. Then, the cells were treated with or without TSD (10 μM) or TNFα, the same as in Section 4.8. Next, western blot and monocyte adhesion assays were performed.

### 4.10. Molecular Modeling Analysis

Molecular docking studies between TSD and each HDAC, HDAC4 (PDB code: 4A69), HDAC5 (PDB code: 5UWI), and HDAC7 (PDB code: 3C10) were performed using Discovery Studio 4.5 software (NeoTrident Technology Ltd., Beijing, China) with the default settings. ChemDraw 14.0 was applied to generate the 3D structures, and LibDock software was also used. The affinity of the receptor-ligand interaction was evaluated based on the LibDock score.

### 4.11. Statistical Analysis

Data analysis was performed using GraphPad Prism 8 software. Data were presented as the mean with SEM. Student’s *t* test or one-way ANOVA was conducted, and *p* < 0.05 was regarded as statistically significant.

## 5. Conclusions

In this study, TSD was isolated from the fermentation of *Streptomyces* sp. CPCC203909 and first reported as a potential activator of KLF2. Furthermore, TSD upregulated KLF2 expression by inhibiting HDAC 4, 5, and 7 and attenuated endothelial inflammation via regulation of the KLF2/NLRP3/Caspase-1/IL-1β signaling pathway.

## Figures and Tables

**Figure 1 ijms-23-13477-f001:**
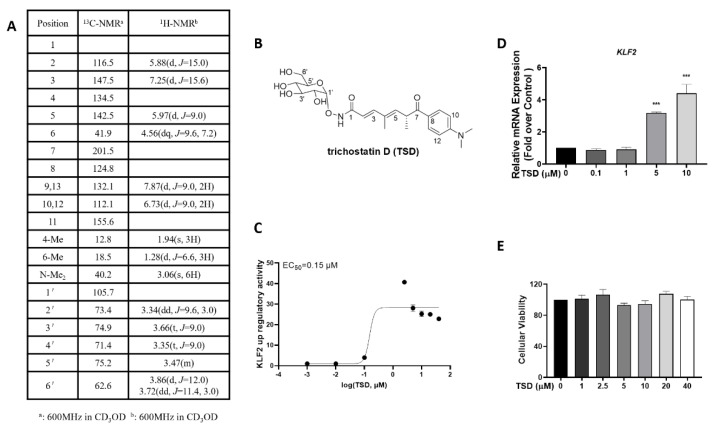
Trichostatin D (TSD) was identified as a novel Krüppel-like factor 2 (KLF2) activator. (**A**) Proton and carbon nuclear magnetic resonance spectroscopy (^1^H-NMR and ^13^C-NMR) data of TSD. (**B**) Chemical structure of TSD. (**C**) KLF2 dose-response curve of TSD. COS-7 cells were transfected with a KLF2-luc promoter-driven luciferase reporter plasmid (KLF2-Luc) for 6 h and then treated with the vehicle (0.1% DMSO) or TSD (0.001, 0.01, 0.1, 2.5, 5, 10, 20, and 40 μM) for 24 h. Then, the activity of the *KLF2* luciferase reporter gene was analyzed. Values are represented as the mean ± SEM; *n* = 3. (**D**) TSD upregulated *KLF2* mRNA levels in human umbilical vein endothelial cell (HUVECs). HUVECs were treated with TSD (0, 0.1, 1, 5, and 10 μM) for 24 h, and then, *KLF2* mRNA levels were detected via real-time-quantitative polymerase chain reaction (RT-qPCR) with *ACTB* (β-actin) as a reference. Values are represented as the mean ± SEM; *n* = 3; one-way ANOVA was used for analysis. *** *p* < 0.0001 vs. 0 μM. (**E**) Cell viability of TSD in HUVECs. HUVECs were inoculated in a 96-well plate and treated with different concentrations of TSD (0, 1, 2.5, 5, 10, 20, and 40 μM) for 24 h and then analyzed with a cell counting kit-8 (CCK-8) kit, as described in the Methods section. The absorbance was measured at 450 nm. Values are represented as the mean ± SEM; *n* = 3; one-way ANOVA was used for analysis.

**Figure 2 ijms-23-13477-f002:**
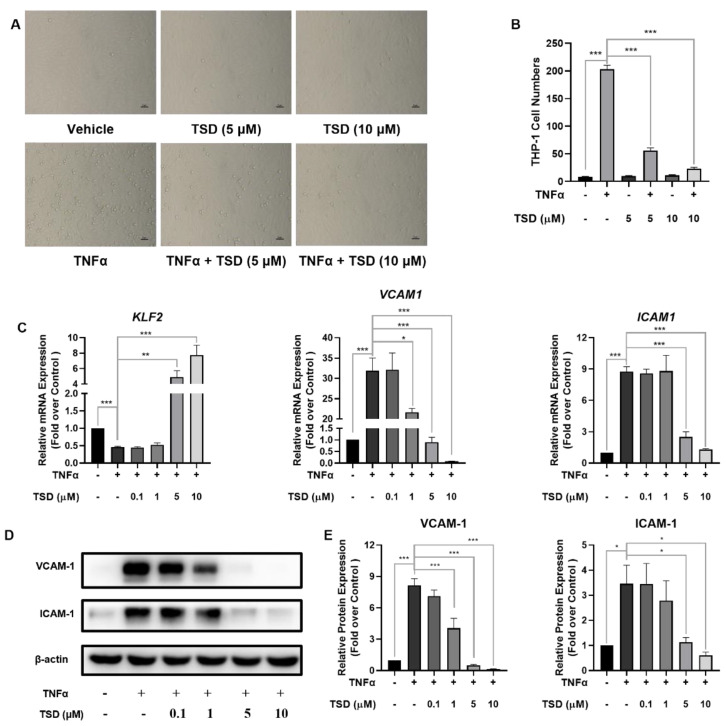
TSD attenuated monocyte adhesion to HUVECs. (**A**,**B**) Representative images (**A**) and quantification results (**B**) of monocyte adhesion to HUVECs after treatment with TSD. The cells were plated in 6-well plates and treated with TSD (5, 10 μM) or DMSO for 18 h; then, tumor necrosis factor alpha (TNFα) (10 ng/mL) was added to the cells for 6 h. THP-1 cells were co-incubated for another 30 min. Values are represented as the mean ± SEM; *n* = 3; one-way ANOVA was used for analysis. *** *p* < 0.0001. (**C**–**E**) mRNA expression (**C**), representative western blot (**D**), and quantification of protein expression of vascular cell adhesion protein 1 (VCAM-1) and intercellular adhesion molecule 1 (ICAM-1) (**E**) after treatment with TSD. The cells were plated in 6-well plates before treatment with TSD (0, 0.1, 1, 5, and 10 μM) for 18 h. TNFα (10 ng/mL) was added to the cells for 6 h, and then, RT-qPCR was performed to assess the mRNA level of *KLF2*, *VCAM1*, and *ICAM1*. Western blot was performed to assess VCAM-1 and ICAM-1 protein levels. The mRNA and protein levels were normalized to *ACTB* (β-actin). Values are represented as the mean ± SEM; *n* = 3; one-way ANOVA was used for analysis. * *p* < 0.05, ** *p* < 0.001, *** *p* < 0.0001.

**Figure 3 ijms-23-13477-f003:**
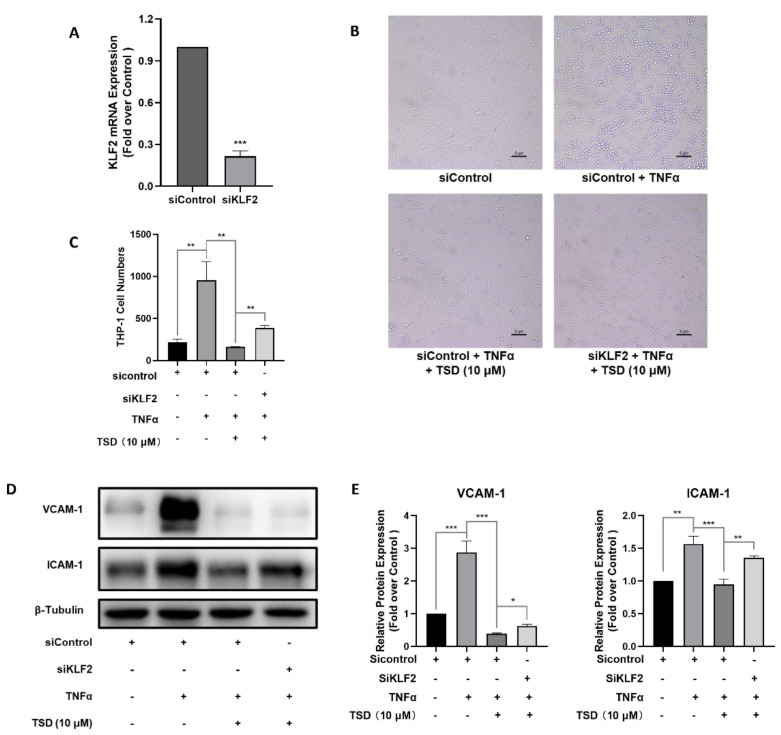
TSD exerts an anti-inflammatory effect dependent on KLF2. (**A**–**E**) Levels of *KLF2* mRNA (**A**), representative monocyte adhesion images (**B**), quantification of monocyte adhesion in HUVECs (**C**), representative western blots (**D**), and quantification of protein expression of VCAM-1 and ICAM-1 (**E**) after treatment with siControl, *KLF2* siRNA (siKLF2), or TSD. HUVECs were cultured in 6-well plates and were transfected with siControl or siKLF2 for 6 h. Then, the medium was replaced with endothelial cell medium and incubated for another 18 h. The cells were then treated with or without TSD (10 μM) for 18 h, followed by 10 ng/mL TNFα for 6 h. Monocyte adhesion assays and western blot analysis were performed, as described in the Methods section. A representative image of each group is shown. The protein levels were normalized to β-tubulin. Values are represented as the mean ± SEM; *n* = 3; Student’s *t* test (**A**) or one-way ANOVA (**C**,**E**) was used for analysis. * *p* < 0.05, ** *p* < 0.001, *** *p* < 0.0001.

**Figure 4 ijms-23-13477-f004:**
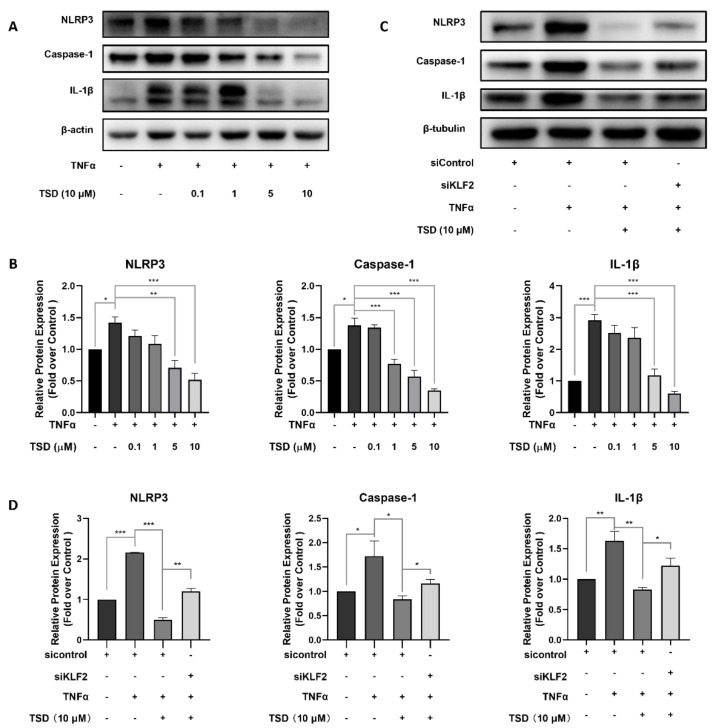
TSD exerts anti-inflammatory effects on HUVECs via inhibition of the NOD-like receptor protein 3 (NLRP3)/Caspase-1/interleukin-1beta (IL-1β) signaling pathway. (**A**,**B**) Representative western blots (**A**) and quantification (**B**) of the protein expression of NLRP3, Casapase-1, and IL-1β after treatment with TSD. Cells in 6-well plates were treated with or without TSD (0, 0.1, 1, 5, and 10 μM) for 18 h, and TNFα (10 ng/mL) was added for 6 h. Western blot was performed to assess NLRP3, Casapase-1, and IL-1β protein levels. (**C**,**D**) Representative western blots (**C**) and quantification (**D**) of the protein expression of NLRP3, Casapase-1, and IL-1β after treatment with siControl and siKLF2. HUVECs were cultured in 6-well plates and transfected with siControl or siKLF2 with Lipofectamine RNAi^MAX^ for 6 h. Then, the medium was replaced with endothelial cell medium and incubated for another 18 h. Then, the cells were treated with or without TSD (10 μM) for 18 h, followed by 10 ng/mL of TNFα for 6 h. Western blot was performed to assess NLRP3, Casapase-1, and IL-1β protein levels. Each group was represented by a representative image. Values are represented as the mean ± SEM; *n* = 3; one-way ANOVA was used for analysis. * *p* < 0.05, ** *p* < 0.001, *** *p* < 0.0001.

**Figure 5 ijms-23-13477-f005:**
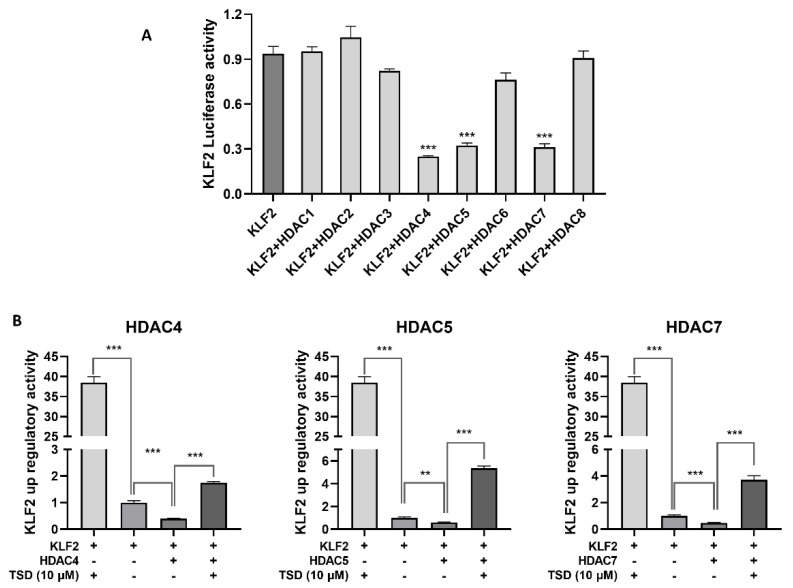
Effect of TSD on histone deacetylase (HDAC)-modulated *KLF2* expression in COS-7 cells. (**A**) Effect of HDACs on *KLF2* expression. COS-7 cells plated in a 96-well plate were co-transfected with HDAC1-8 plasmids and KLF2 plasmids (20 ng HDAC plasmids and 180 ng *KLF2* luciferase reporter plasmids per well) for 6 h, and the medium was replaced with DMEM and incubated for another 18 h. Then, the activity of the *KLF2* luciferase reporter gene was detected. (**B**) Effect of TSD on HDACs and KLF2. COS-7 cells plated in a 96-well plate were co-transfected with HDAC and KLF2 plasmids (20 ng HDACs and 180 ng KLF2 per well) for 6 h and then treated with TSD (10 μM) for 18 h. The activity of the *KLF2* luciferase reporter gene was measured. (**A**,**B**) Values are represented as the mean ± SEM; *n* = 4; Student’s *t* test was used for analysis. ** *p* < 0.001, *** *p* < 0.0001.

**Figure 6 ijms-23-13477-f006:**
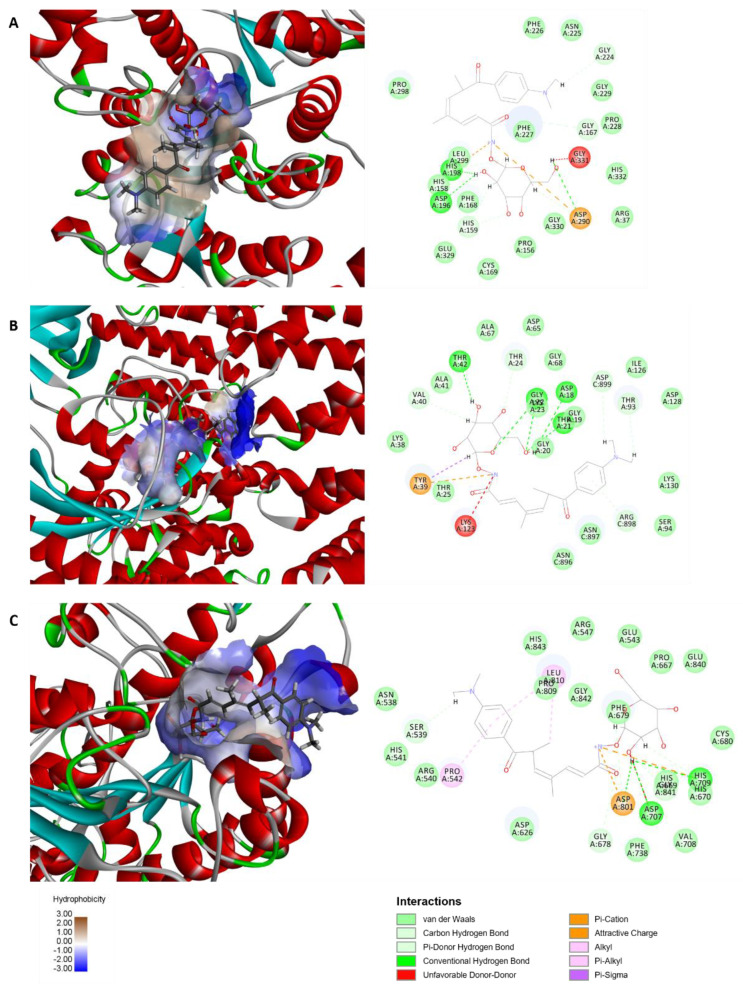
Molecular docking analysis between HDAC4, HDAC5, and HDAC7 proteins with TSD by Discovery Studio 4.5. 3D and 2D modes of the interaction pocket between (**A**) TSD with HDAC4, (**B**) TSD with HDAC5, and (**C**) TSD with HDAC7.

**Figure 7 ijms-23-13477-f007:**
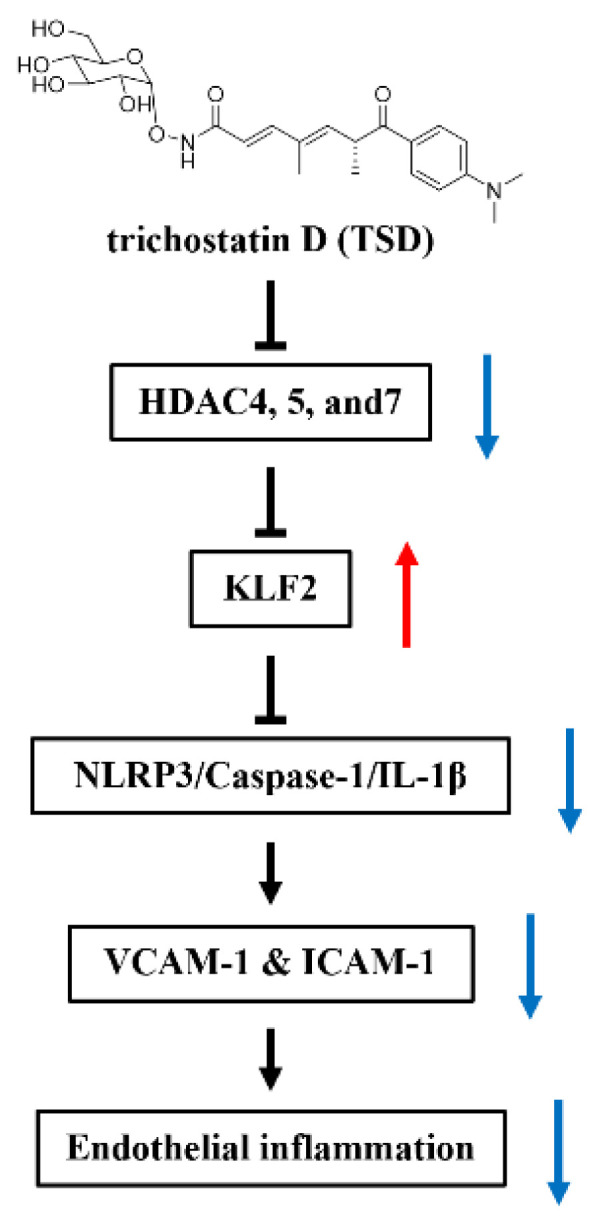
The main signal pathway by which TSD inhibits inflammation in HUVECs. The red arrow means up-regualtion, and the blue arrow means down-regulation.

## Data Availability

Data available on request from the authors.

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
