# Peer review of "Trichostatin D as a Novel KLF2 Activator Attenuates TNFα-Induced Endothelial Inflammation"

_ijms, 2022, doi:10.3390/ijms232113477_

Round 1

Reviewer 1 Report

These investigators screened microbial sources for agonists of KLF2, an anti-inflammatory endogenous factor that has anti-atherogenic properties.  They isolated an analogue of trichostatin A, termed trichostatin D (TSD), from a Streptomyces species and demonstrated that it was a dose-dependent KLF2 activator in human umbilical cord endothelial cells (HUVEC).  They showed downstream TSD inhibition of monocyte adherence to the cells and VCAM-1 and ICAM-1 expression by the cells after TNF-alpha treatment and that the KLF2 stimulation was mediated by inhibition of HDAC 4, 5 and 7.  This is a well-written summary of a sound in vitro study.  I cannot find any place in the manuscript where HDAC is defined.  There should be a list or table of abbreviations.  In the Materials and Methods section, the authors should specify the semi-preparative HPLC column that they used and the source and nature of the CCK-8 cell enumeration kit.

Author Response

Response: Thank you for your suggestions.

1) HDAC is defined in the abstract and introduction part in the revised manuscript.   

Abstract: “Furthermore, histone deacetylase (HDAC) overexpression and molecular docking analysis results showed that TSD upregulated KLF2 expression by inhibiting HDAC 4, 5, and 7 activity.”

Introduction: “Furthermore, we demonstrated that TSD upregulated KLF2 expression by inhibiting histone deacetylase (HDAC) 4, 5, and 7 and attenuated endothelial inflammation through regulation of the KLF2/NLRP3/Caspase-1/IL-1β signaling pathway.”

2) We carefully examined all the abbreviations to make sure all of them were listed in the revised manuscript, so we don’t add the list of abbreviations in this article.

3) We added the information of column (X Charge C18 5 μm, Acchrom, Beijing, China) in the revised manuscript.

4) We added the source and nature of the CCK-8 kit in the materials and methods section “4.5. Cell Viability Assay” as follow.

“A sensitive colorimetric assay was performed to quantify the viable cells in cytotoxicity assay with a Cell Counting Kit-8 (CCK-8, Applygen, Beijing, China) using a microplate reader.”

Reviewer 2 Report

I have reviewed the manuscript “Trichostatin D as a novel KLF2 activator attenuates TNFα-induced endothelial inflammation” by Lei et al submitted for publication in the IJMS (MDPI).

In the study, the authors aimed to identify and study the Trichostatin D (TSD), a metabolite from Streptomyces which could be a potential KLF2 activator. In the study, they used various methods to prove the TLD as the novel KLF2 activator which attenuates TNF alpha-induced endothelial inflammation.

The overall study, methods, and experiments are well-designed and the results were analyzed with appropriate controls and statistical methods.  But has some minor questions and concerns to address.

As statins are studied well and proven to attenuate endothelial dysfunction via KLF2 activation. Why did the authors not compare the effects of two major statins (Atorvastatin and Simvastatin) with TSD to compare the effectiveness of TSD with statins?

 This will be very important to show and prove TSD is a novel drug. If statins exhibit the same effect as TSD then the whole aim of the study is not fulfilled hence there is already an existing clinically proven drug in the market available to achieve the same therapeutic effect.

 I recommend the authors perform the experiments to compare the efficiency of TSD with statins and add it to the revised manuscript.

Author Response

Response: Thank you for your constructive suggestions. We try our best to solve your concerns by providing more evidences following your suggestions in the revised manuscript.

We agreed that statins are studied well and proven to attenuate endothelial dysfunction via KLF2 activation. Therefore, we compared the up-regulating KLF2 and anti-inflammatory effects of TSD with statins including Simvastatin and Atorvastatin in the revised manuscript, and these results were added in the discussion part.

In addition, as statins are studied well and proven to attenuate endothelial dysfunction via KLF2 activation, we then performed KLF2 promoter luciferase assays in COS-7 cells and TNFα-induced monocyte adhesion assays to compare the effects of statins (Simvastatin and Atorvastatin) with TSD. As shown in Supplementary data 1A-B, Simvastatin maximally increased the luciferase activity of KLF2 by about 8-fold, with an EC50 value of 1.74 μM (Supplementary data 1A), and Atorvastatin maximally increased the luciferase activity of KLF2 by about 12-fold, with an EC50 value of 5.14 μM (Supplementary data 1B). Monocyte adhesion assays results showed that Simvastatin (1 and 10 μM) and Atorvastatin (1 and 10 μM) treatment attenuated TNFα-induced monocyte adhesion (Supplementary data 2) and exhibited anti-inflammatory effects in endothelial cells (Supplementary data 2). Furthermore, our results indicated that TSD is more effective in upregulating KLF2 expression and anti-inflammatory effects than Simvastatin and Atorvastatin according to the results of KLF2 promoter luciferase assay and inhibiting monocyte adhesion (Supplementary data 1&2).

Round 2

Reviewer 2 Report

Dear Authors,

I have reviewed the resubmitted manuscript after incorporating the experiments with Simvastatin and Atorvastatin per my review suggestions.

Interestingly, TSD shows better efficiency than Simvastatin and Atorvastatin, this makes the manuscript more intriguing and interesting to the readers.

I recommend the editorial board, know the manuscript is suitable and may be accepted for publication in the present format.